# KNOWLEDGE DISTILLATION FOR CLOSED-SOURCE LANGUAGE MODELS

## ABSTRACT

Closed-source language models such as GPT-4 have achieved remarkable performance. Many recent studies focus on enhancing the capabilities of smaller models through knowledge distillation from closed-source language models. However, due to the incapability to directly access the weights, hidden states, and output distributions of these closed-source models, the distillation can only be performed by fine-tuning smaller models with samples generated by closed-source language models, which constrains the effectiveness of knowledge distillation. In this paper, we propose to estimate the output distributions of closed-source language models within a Bayesian estimation framework, involving both prior and posterior estimation. The prior estimation aims to derive a prior distribution by utilizing the corpus generated by closed-source language models, while the posterior estimation employs a proxy model to update the prior distribution and derive a posterior distribution. By leveraging the estimated output distribution of closed-source language models, traditional knowledge distillation can be executed. Experimental results demonstrate that our method surpasses the performance of current models directly fine-tuned on data generated by closed-source language models.

## 1 INTRODUCTION

While closed-source large language models (LLMs) such as GPT-3.5 and GPT-4 have shown great superiority over open-source counterparts like LLaMA (Touvron et al., 2023) and Falcon (Penedo et al., 2023), they can only be accessed via API calls and allow limited customization and transparency. One way to address this problem is to transfer their capabilities to open-source language models, typically smaller in size, by prompting closed-source LLMs to generate samples that reflect their capabilities and fine-tuning open-source language models on these samples (Hsieh et al., 2023; Jiang et al., 2023; Ho et al., 2022). However, this approach only enables open-source language models to emulate the input-output behavior of closed-source LLMs without acquiring their intrinsic knowledge related to logits, weights, activations, and so forth.

Knowledge distillation (KD) (Hinton et al., 2015) is a popular compression technology that aims to train a small but strong student model by distilling knowledge from a large teacher model. Among various sources of knowledge, the logits of the teacher model are typically utilized as an essential part of the objective function, implemented by minimizing the Kullback-Leibler (KL) divergence between the output distribution (soft labels) of the teacher model and the output distribution of the student model. This approach enables the student model to mimic the predictive behavior and acquire the knowledge of the teacher model. However, such approaches are not readily applicable to closed-source LLMs as the soft labels are not feasible.

To tackle this challenge, we propose to estimate the output distributions of closed-source LLMs within a Bayesian estimation framework, including both prior and posterior estimation. The aim of prior estimation is to derive a prior distribution by leveraging the corpus generated by closed-source language models. The rationale is that the corpus may contain coarse-grained information regarding the output distributions of closed-source LLMs. Meanwhile, the posterior estimation utilizes a proxy model, another open-source LLM typically larger than the student model, to calibrate the results of the prior estimation. This proxy model is initially aligned with the closed-source teacher model and then functions as a bridge between the teacher and the student, as illustrated in Figure 1. By leveraging the estimated output distribution of closed-source LLMs, traditional knowledge distillation can

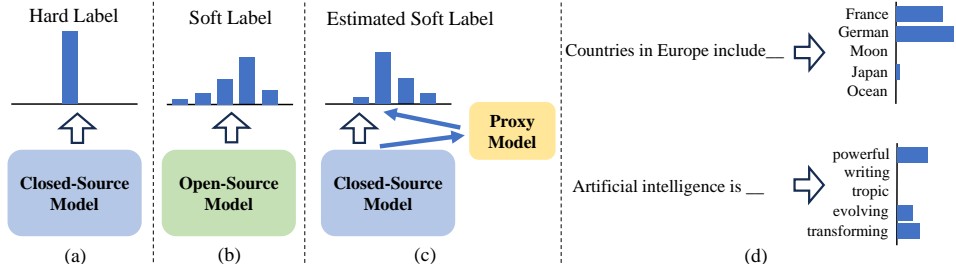

Figure 1: (a) In current knowledge distillation of closed-source models, only hard labels can be obtained. (b) In traditional knowledge distillation of open-source models, soft labels can be obtained. (c) Our method obtains estimated soft labels from closed-source models by leveraging a proxy model. (d) Compared to hard labels, soft labels allow students to learn more profound knowledge by guiding them to learn from multiple valid targets during distillation.

be carried out. Compared to previous approaches addressing this objective, our method enables the student model to learn from both the generated samples by the closed-source teacher and the soft labels provided by the proxy model, allowing the distillation of more intrinsic knowledge.

To validate our approach, we performed comprehensive experiments on a range of well-established benchmarks, including complex reasoning datasets BBH (Suzgun et al., 2022) and ARC Clark et al. (2018), knowledge-based datasets AGIEval (Zhong et al., 2023) and MMLU (Hendrycks et al., 2021), commonsense reasoning dataset CSQA (Talmor et al., 2019), and mathematical reasoning dataset GSM8K (Cobbe et al., 2021). We used GPT-4 as the closed-source teacher model, LLaMA-33B as the proxy model, and LLaMA-13B/7B as the student model. The empirical results demonstrate the superiority of our method over directly fine-tuning the student model on samples generated by GPT-4, with an average improvement from points 36.31 to 39.43 across the six benchmarks. The experimental results show that, the introduction of a proxy model can serve as an intermediary bridge for student model to learn knowledge from the closed-source teacher model. It benefits from the proxy model that aligns better with the teacher model. This facilitates the transfer of more profound knowledge from the closed-source teacher model to the student model more effectively.

## 2 RELATED WORK

The concept of knowledge distillation (KD) was originally introduced by Hinton et al. (2015) with the aim of transferring the knowledge from a teacher model to a smaller student model. Current KD methods can be organized into two primary categories: knowledge distillation for open-source models and knowledge distillation for closed-source models.

### 2.1 OPEN-SOURCE KNOWLEDGE DISTILLATION

KD can be applied to open-source models for natural language understanding. For instance, Sanh et al. (2019) applied KD to the pre-training process of BERT (Devlin et al., 2019), yielding smaller models with minor performance drops. Jiao et al. (2020) allowed the student model's intermediate features to mimic the teacher model's intermediate features, by minimizing the Mean Squared Error (MSE) loss function. KD can also be applied to open-source models for natural language generation. Lin et al. (2020) investigated the exposure bias problem in the process of distillation for open-source language models. Similarly, Agarwal et al. (2023) studied the distribution mismatch between output sequences during training and the sequences generated by the open-source student during its deployment. Other approaches, such as the one proposed by Gu et al. (2023), focused on distilling open-source LLMs like LLaMA (Touvron et al., 2023). However, in all these methods, the student model needs access to the internal weights and features of the teacher model, which is not feasible in the context of distilling closed-source LLMs.

Most similar to our work, Mirzadeh et al. (2019) introduced an intermediate network to bridge the parameter size gap between the CNN teacher model and the CNN student model. In contrast to their approach, we introduce an intermediate network with the specific purpose of estimating output distributions of closed-source LLMs and achieving enhanced knowledge distillation.

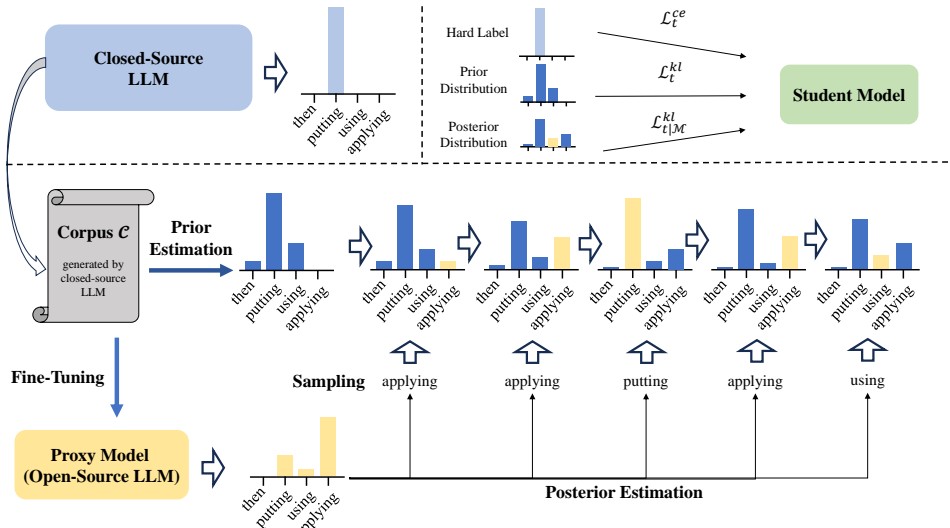

Figure 2: Overview of our method. The output distributions of closed-source LLMs are estimated within a Bayesian estimation framework, including both prior and posterior estimation. The prior estimation leverages the corpus generated by closed-source language models to derive a prior distribution, while the posterior estimation utilizes a proxy model to calibrate the results of the prior estimation. Traditional knowledge distillation is applied using the estimated output distributions.

## 2.2 CLOSED-SOURCE KNOWLEDGE DISTILLATION

In light of the remarkable performance of closed-source LLMs such as GPT-3.5 and GPT-4, numerous studies have shifted their attention toward transferring the diverse capabilities from these proprietary LLMs into smaller open-source models. For instance, Liang et al. (2023) improved the mathematical capability of a small model by training it with tailored exercise samples generated by GPT-3 (Brown et al., 2020). To transfer the code generation capability, Azerbayev et al. (2023) prompted Codex (Chen et al., 2021) to create natural language-code pairs and fine-tuned a smaller model on those samples. To transfer the tool usage capability, Gou et al. (2023) utilized GPT-4 to generate interactive tool-use trajectories as training samples for the target model. Other approaches, such as Hsieh et al. (2023); Ho et al. (2022); Mukherjee et al. (2023) utilized rationales generated by closed-source LLMs as training data to transfer their general reasoning capabilities.

To sum up, these works typically transfer the capabilities of closed-source LLMs by prompting them to generate samples, which are then utilized to train a smaller open-source model. Essentially, these approaches mainly capture the input-output patterns of closed-source LLMs without delving into more nuanced knowledge as traditional knowledge distillation methods. In contrast, our approach aims to estimate the output distribution of closed-source LLMs to train the student model within the traditional knowledge distillation framework.

## 3 METHOD

To perform knowledge distillation in traditional approaches, we propose to estimate the output distributions of closed-source LLMs within a Bayesian estimation framework, which includes both prior and posterior estimation. For a specific text input, prior estimation leverages the corpus generated by closed-source language models to derive an initial approximation for the distribution of the output. Meanwhile, posterior estimation relies on another open-source LLM as a proxy to fine-tune the results of prior estimation. This proxy model serves as a bridge between the teacher (closed-source) and the student (open-source) models, as illustrated in Figure 2. Therefore, the proxy model is selected to be a larger language model than the student model and is initially aligned with the closed-source teacher model using the aforementioned corpus. Finally, we perform knowledge distillation using the estimated output distributions of the closed-source teacher LLM.

| Notations | Descriptions |
|-----------|--------------|
| $\mathcal{T}$ | Closed-source teacher model |
| $\mathcal{S}$ | Open-source student model |
| $\mathcal{M}$ | Open-source proxy model |
| $Y$ | Output token sequence |
| $X$ | Input token sequence |
| $p_{Y_t}$ | Probability $\Pr(Y_t|X, Y_{<t})$ given by $\mathcal{T}$ |
| $q_{Y_t}$ | Probability $\Pr(Y_t|X, Y_{<t})$ given by $\mathcal{S}$ |
| $P_{Y_t}$ | Discrete random variable associated with the value of $p_{Y_t}$ |

Table 1: Main notations and descriptions.

## 3.1 PROBLEM STATEMENT

In this section, we first introduce the objective function in traditional knowledge distillation for language models. We use $\mathcal{T}$ and $\mathcal{S}$ to represent the closed-source teacher model and open-source student model, respectively. Let $X$ denote the input sequence of tokens and $Y$ denote the output sequence of tokens. At time $t$, the probability of generating an output token $Y_t$ can be represented as $\Pr(Y_t|X, Y_{<t})$. Let $p_{Y_t}$ be the probability $\Pr(Y_t|X, Y_{<t})$ given by $\mathcal{T}$, let $q_{Y_t}$ be the probability $\Pr(Y_t|X, Y_{<t})$ given by $\mathcal{S}$. Let $\mathbb{1}_{Y_t}$ be the one-hot encoded label at time $t$ provided by $\mathcal{T}$. The traditional token-level objective function of knowledge distillation at time $t$ be derived as follows:

$$\mathcal{L}_t^{\text{traditional}} = -\sum_{w \in \mathbb{V}} \mathbb{1}_{Y_t=w} \log q_{Y_t=w} + \sum_{w \in \mathbb{V}} p_{Y_t=w} \log \frac{p_{Y_t=w}}{q_{Y_t=w}}, \tag{1}$$

where $\mathbb{V}$ is the vocabulary, $w$ is a token in the vocabulary. $\mathcal{L}_t^{\text{traditional}}$ consists of two terms: the first term involves computing cross-entropy loss with hard labels, and the second term involves computing KL loss with soft labels. In the context of knowledge distillation of $\mathcal{T}$, the second term is typically omitted because obtaining $p_{Y_t}$ is not directly feasible.

## 3.2 ESTIMATION METHODS

In this section, we elaborate on the proposed estimation methods: prior estimation and posterior estimation. Both methods are designed to estimate the soft labels (i.e., $p_{Y_t}$) of $\mathcal{T}$.

### 3.2.1 PRIOR ESTIMATION

The prior estimation aims to obtain a coarse-grained $\hat{p}_{Y_t}$ to approximate $p_{Y_t}$ at each time step $t$. The method achieves this by leveraging a corpus $\mathcal{C}$ generated by $\mathcal{T}$, through an optimized n-gram algorithm. Given a specific output token sequence $Y_{\leq t} \in \mathcal{C}$, assuming $Y_t = w_t$, where $w_t$ is a specific token in $\mathbb{V}$. For those tokens $w \in \mathbb{V}$, if $w = w_t$:

$$\hat{p}_{Y_t=w} = \frac{\#(Y_t = w, Y_{t-1} = w_{t-1}, \ldots, Y_{t-n} = w_{t-n})}{\gamma \#(Y_{t-1} = w_{t-1}, \ldots, Y_{t-n} = w_{t-n})} + \frac{\gamma - 1}{\gamma}, \tag{2}$$

otherwise:

$$\hat{p}_{Y_t=w} = \frac{\#(Y_t = w, Y_{t-1} = w_{t-1}, \ldots, Y_{t-n} = w_{t-n})}{\gamma \#(Y_{t-1} = w_{t-1}, \ldots, Y_{t-n} = w_{t-n})}, \tag{3}$$

where the # represents the count of a specific output token sequence appears in $\mathcal{C}$. The $n$ is the window size. The $\gamma$ is a hyperparameter, $\gamma \in \mathbb{Z}^+$. The $\gamma$ is used to adjust dominant probability contribution of the token $w_t$. For instance, when $\gamma = 2$, term $\frac{\gamma-1}{\gamma}$ ensures that the probability $\hat{p}_{Y_t=w_t}$ is greater than 50%. An assumption behind the prior estimation is that $\mathcal{T}$ typically generates the next token with a strong association to the most recent preceding tokens. Through Equation 2 and 3, we obtain an initial estimate $\hat{p}_{Y_t}$ for the soft labels $p_{Y_t}$. We refer to $\hat{p}_{Y_t}$ as the prior distribution.

### 3.2.2 POSTERIOR ESTIMATION

The prior distribution $\hat{p}_{Y_t}$ serves as a coarse-grained approximation for $p_{Y_t}$. To further refine the prior distribution and get a better approximation for $p_{Y_t}$, we introduce posterior estimation. The posterior estimation is primarily achieved by introducing a proxy $\mathcal{M}$ of $\mathcal{T}$ (typically an open-source LLM with a larger size than $\mathcal{S}$) under the Bayesian estimation framework. This estimation involves continuously sampling from $\mathcal{M}$ to refine the prior distribution. The $\mathcal{M}$ is previously fine-tuned on the corpus $\mathcal{C}$ generated by $\mathcal{T}$ for preliminary alignment with $\mathcal{T}$. The motivation behind introducing $\mathcal{M}$ is to leverage it as a bridge between the closed-source teacher $\mathcal{T}$ and the open-source student $\mathcal{S}$, serving a purpose of better estimating the soft labels $p_{Y_t}$ of $\mathcal{T}$.

We consider the value of $p_{Y_t}$ can be described by a discrete random variable denoted as $P_{Y_t}$ (the transformation to continuous case is straightforward, but we discuss the discrete case for better understanding.). We define $P_{Y_t}$ with $m$ possible discrete values $p^1, p^2, \ldots, p^m$, where $p^1, p^2, \ldots, p^m$ form a number sequence increasing by $1/m$ from 0 to 1 (e.g., $0.00, 0.01, 0.02, \ldots, 0.99$, with $m = 100$). According to the prior distribution $\hat{p}_{Y_t}$, the probability mass function (PMF) $\Pr(P_{Y_t} = p^i)$ of $P_{Y_t}$ can be predefined in a way that satisfies the following constraint:

$$\mathbb{E}(P_{Y_t}) = \sum_{i=1}^{m} p^i \Pr(P_{Y_t} = p^i) = \hat{p}_{Y_t} \tag{4}$$

Equation 4 implies that the PMF can vary, as long as the expectation $\mathbb{E}(P_{Y_t})$ equals $\hat{p}_{Y_t}$. In practice, $m$ should be sufficiently large (e.g., $m = 100$). Calibrating the prior distribution involves updating the PMF through sampling from $\mathcal{M}$. We feed $X$ and $Y_{<t}$ into $\mathcal{M}$, a token $\hat{w} \in \mathbb{V}$ is sampled at time $t$. Given $\hat{w}$, and a token $w \in \mathbb{V}$, event $A$ is defined as follows: if $w = \hat{w}$, $A = 1$; otherwise, $A = 0$. In a sampling round, we update the PMF $\Pr(P_{Y_t} = p^i)$ based on the event $A$. If event $A = 1$ occurs, according to Bayes' theorem:

$$\Pr(P_{Y_t=w} = p^i | A = 1) \propto \Pr(A = 1 | P_{Y_t=w} = p^i) \Pr(P_{Y_t=w} = p^i) = p^i \Pr(P_{Y_t=w} = p^i), \tag{5}$$

where $w \in \mathbb{V}$, $i \in \{1, 2, \ldots, m\}$. We get a normalization factor $\eta$ by:

$$\eta = \sum_{i=1}^{m} p^i \Pr(P_{Y_t=w} = p^i) \tag{6}$$

Then the value of $\Pr(P_{Y_t=w} = p^i | A = 1)$ can be calculated as $\frac{1}{\eta} p^i \Pr(P_{Y_t=w} = p^i)$. If event $A = 0$ occurs instead, according to Bayes' theorem:

$$\Pr(P_{Y_t=w} = p^i | A = 0) \propto \Pr(A = 0 | P_{Y_t=w} = p^i) \Pr(P_{Y_t=w} = p^i) = (1 - p^i) \Pr(P_{Y_t=w} = p^i), \tag{7}$$

where $w \in \mathbb{V}$, $i \in \{1, 2, \ldots, m\}$. We get a normalization factor $\eta$ by:

$$\eta = \sum_{i=1}^{m} (1 - p^i) \Pr(P_{Y_t=w} = p^i) \tag{8}$$

Then the value of $\Pr(P_{Y_t=w} = p^i | A = 0)$ can be calculated as $\frac{1}{\eta}(1 - p^i) \Pr(P_{Y_t=w} = p^i)$. At this point, one sampling iteration concludes. The prior $\Pr(P_{Y_t} = p^i)$ will be replaced by the posterior $\Pr(P_{Y_t} = p^i | A = 1)$ or $\Pr(P_{Y_t} = p^i | A = 0)$ in the next iteration. After multiple rounds of sampling from $\mathcal{M}$, we denote the final PMF as $\Pr(P_{Y_t} = p^i | \mathcal{M})$. The $p_{Y_t}$ can be approximated by calculating the conditional expectation as follow:

$$\mathbb{E}(P_{Y_t} | \mathcal{M}) = \sum_{i=1}^{m} p^i \Pr(P_{Y_t} = p^i | \mathcal{M}) \tag{9}$$

We refer $\mathbb{E}(P_{Y_t} | \mathcal{M})$ to as the posterior distribution.

## 3.3 Overall Objective

The overall objective function at time step $t$ comprises three objectives. Let $\mathbb{1}_{Y_t}$ be the one-hot encoded label provided by $\mathcal{T}$, the first objective at time step $t$ can be derived by calculating the cross-entropy loss as $\mathcal{L}_t^{\text{ce}} = -\sum_{w \in \mathbb{V}} \mathbb{1}_{Y_t = w} \log q_{Y_t = w}$. The second objective at time step $t$ can be derived based on the prior distribution as $\mathcal{L}_t^{\text{kl}} = \sum_{w \in \mathbb{V}} \hat{p}_{Y_t = w} \log \frac{\hat{p}_{Y_t = w}}{q_{Y_t = w}}$. We first normalize $\mathbb{E}(P_{Y_t}|\mathcal{M}) = \frac{\mathbb{E}(P_{Y_t}|\mathcal{M})}{\sum_{w \in \mathbb{V}} \mathbb{E}(P_{Y_t = w}|\mathcal{M})}$, then the third objective at time step $t$ can be derived based on the posterior distribution as $\mathcal{L}_{t|\mathcal{M}}^{\text{kl}} = \sum_{w \in \mathbb{V}} \mathbb{E}(P_{Y_t = w}|\mathcal{M}) \log \frac{\mathbb{E}(P_{Y_t = w}|\mathcal{M})}{q_{Y_t = w}}$. Given an output tokens sequence with length $T$, the overall objective function can be derived as follows:

$$\mathcal{L} = \frac{1}{T} \sum_{t=1}^{T} (\mathcal{L}_t^{\text{ce}} + \alpha \mathcal{L}_t^{\text{kl}} + \beta \mathcal{L}_{t|\mathcal{M}}^{\text{kl}}) \tag{10}$$

Where the $\alpha$ and $\beta$ are hyperparameters used to adjust the contributions of the $\mathcal{L}_t^{\text{kl}}$ and $\mathcal{L}_{t|\mathcal{M}}^{\text{kl}}$ in the total loss. When $\alpha > 0$ and $\beta = 0$, $\mathcal{L}$ becomes the loss for prior distillation. When $\alpha = 0$ and $\beta > 0$, $\mathcal{L}$ becomes the loss for posterior distillation.

## 4 Experimental Setup

In this section, we conduct a series of experiments to validate the effectiveness of our method.

### 4.1 Datasets

We mainly utilize the OpenOrca (Mukherjee et al., 2023) dataset as our training corpus. The OpenOrca dataset was created by prompting closed-source LLMs, such as GPT-4, with diverse inputs and collecting the corresponding output sequences. We follow the settings in OpenOrca-Preview1-13B[1] of paper Mukherjee et al. (2023). We also utilize the Alpaca (Taori et al., 2023) dataset as the training corpus. The Alpaca dataset was generated by providing diverse inputs to the closed-source LLM text-davinci-003 prompt and collecting the corresponding output sequences.

For evaluation, we utilize benchmarks including complex reasoning datasets BBH (Suzgun et al., 2022) and ARC Clark et al. (2018), knowledge-based datasets AGIEval (Zhong et al., 2023) and MMLU (Hendrycks et al., 2021), commonsense reasoning dataset CSQA (Talmor et al., 2019), and mathematical reasoning dataset GSM8K (Cobbe et al., 2021). These benchmarks assess the model across wide range of capabilities including reading comprehension, commonsense knowledge, mathematical skills and logical reasoning. Following the settings of Mukherjee et al. (2023), aside from GSM8K, we focus on tasks that involve multiple-choice questions.

### 4.2 Backbone Models

We employ currently state-of-the-art closed-source LLMs GPT-4 as well as text-davinci-003 as the closed-source teacher models. We utilize LLaMA-7B and LLaMA-13B as student models, which are initialized with pre-trained weights obtained from Hugging Face[2]. We choose LLaMA-33B as the proxy model. We employ top-p sampling for decoding. We train our models on 8 32GB V100 GPUs. Additional details can be found in Appendix A.

### 4.3 Baselines

We consider instruction fine-tuning (IFT) approach as our baseline. IFT involves fine-tuning the student model on the samples generated by the teacher model without using soft labels. We implement the baseline models of our own version ourselves. We implement our own version of baseline models. To ensure a fair comparison with other baseline models, we exclusively include models that have access to their original fine-tuning datasets. As a result, our chosen baseline models are

---

[1] https://huggingface.co/Open-Orca/OpenOrca-Preview1-13B
[2] https://huggingface.co/models

| Models | #Params | BBH | AGIEval | ARC | MMLU | CSQA | GSM8K | Average |
|---|---|---|---|---|---|---|---|---|
| GPT-4 | - | 67.4 | 56.4 | - | 86.4 | - | 92.0 | - |
| LLaMA-7B (IFT) | 7B | 36.08 | 24.14 | 47.49 | 38.81 | 58.71 | 12.65 | 36.31 |
| LLaMA-7B (ours) | 7B | 38.52 | 26.92 | 52.40 | 41.18 | 62.52 | 14.97 | **39.43** |
| OpenOrca-Preview1-13B | 13B | 41.47 | 30.12 | 59.77 | 48.10 | 69.77 | 18.22 | 44.58 |
| LLaMA-13B (IFT) | 13B | 42.77 | 26.74 | 58.2 | 45.3 | 66.27 | 20.93 | 43.37 |
| LLaMA-13B (ours) | 13B | 44.83 | 29.35 | 61.84 | 48.17 | 68.94 | 23.36 | **46.08** |

Table 2: The results of the LLaMA models with different sizes on six benchmarks. We compare our approach to methods directly instruction fine-tuning on the hard labels. The performance of OpenOrca-Preview1-13B is assessed through our own evaluation. All student models are trained on the OpenOrca dataset.

| Models | #Params | BBH | AGIEval | ARC | MMLU | CSQA | GSM8K | Average |
|---|---|---|---|---|---|---|---|---|
| text-davinci-003 | - | 70.7 | 41.9 | - | 64.6 | - | - | - |
| Alpaca-7B | 7B | 34.19 | 24.16 | 39.35 | 33.66 | 36.16 | 13.99 | 30.25 |
| LLaMA-7B (ours) | 7B | 34.92 | 24.32 | 40.3 | 34.14 | 38.32 | 14.33 | **31.06** |
| Alpaca-13B | 13B | 38.1 | 26.9 | 52.57 | 41.41 | 55.27 | 19.27 | 38.92 |
| LLaMA-13B (ours) | 13B | 40.82 | 28.35 | 53.84 | 42.17 | 56.78 | 19.83 | **40.3** |

Table 3: The results of the LLaMA models with different sizes on six benchmarks. We compare our method with Alpaca. All student models are trained on the Alpaca dataset.

OpenOrca-Perview1-13B from Mukherjee et al. (2023) and Alpaca (Taori et al., 2023), which have been fine-tuned on the samples generated by the teacher model.

## 5 RESULT AND ANALYSIS

In this section, we present the main results, ablation studies and additional experiments. All corpus for proxy model fine-tuning, prior estimation, posterior estimation, and student distillation are identical. Unless otherwise specified, the default training corpus we utilize is OpenOrca.

### 5.1 MAIN RESULTS

Table 2 shows the performance comparison of our method against baseline models. Detailed experimental results can be found in Appendix C. The training corpus we utilized in Table 2 is the OpenOrca dataset. Our method outperforms OpenOrca-Preview1-13B from Mukherjee et al. (2023) as well as our own implemented IFT models under both the 7B and 13B model parameter scales. Table 3 shows the performance comparison between our method and Alpaca. The training corpus we utilized in Table 3 is the Alpaca dataset. It shows a similar trend, with our method outperforming Alpaca. A case study in Table 4 demonstrates that our model exhibits better comprehension and answer generation capabilities in terms of reasoning ability compared to the baseline IFT. The experimental results not only demonstrate the effectiveness of our approach for both 7B and 13B student model scales but also validate the effectiveness of using estimated soft labels.

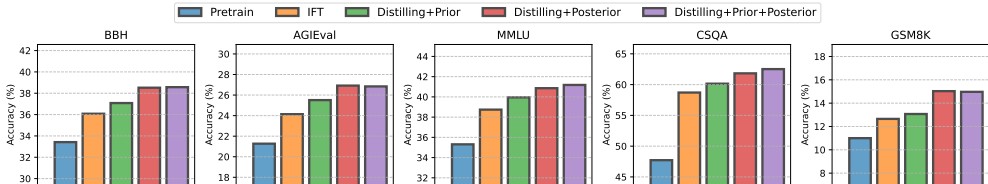

Figure 3: Ablation results of LLaMA-7B across five representative benchmarks. In Distilling+Prior we adjust $\alpha = 0.5, \beta = 0$, to investigate the effect of the prior estimation. In Distilling+Posterior we adjust $\alpha = 0, \beta = 1$, to investigate the effect of the posterior estimation. In Distilling+Prior+Posterior we adjust $\alpha = 0.5, \beta = 1$ to study combined effect of both.

| Instruction | The fastest car was 1.25 times faster than the 2nd fastest car. The 2nd fastest car was 5 times faster than the average car. The average car can go 120 Miles per hour. How fast does the faster car go? |
|---|---|
| Ours | The 2nd fastest car is 5 times faster than the average car, so it can go 5 x 120 = 600 Miles per hour. The faster car is 1.25 times faster than the 2nd fastest car, so it can go 1.25 x 600 = 750 Miles per hour. 
 Therefore, the faster car goes 750 Miles per hour. ✓ |
| IFT | The faster car can go 1.25 x 5 = 60 Miles per hour. 
 Therefore, the faster car goes 60 Miles per hour. ✗ |

Table 4: Examples from GSM8K dataset. Our method enhances the student model's comprehension and leads to more detailed responses compared to the IFT method.

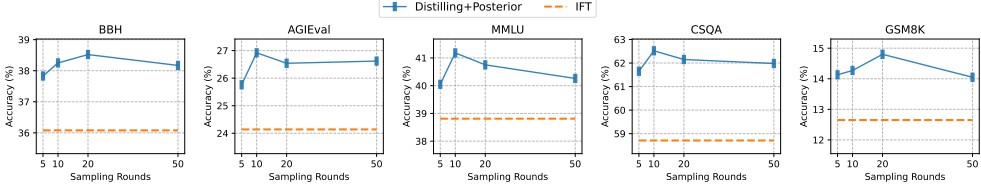

Figure 4: Comparing the performance of knowledge distillation utilizing the posterior distribution under various sampling round configurations with IFT, employing the model with LLaMA-7B.

## 5.2 ABLATION STUDY

This ablation study examines the impact of components within our method. While retaining the standard cross-entropy loss, we evaluate the effect of the prior estimation, and the posterior estimation. All results are presented in Figure 3.

**Effect of the prior estimation** Retaining the cross-entropy loss, we incorporate the KL loss involving the prior distribution for training. This training method is denoted as Distilling+Prior. As shown in Figure 3, Distilling+Prior consistently outperforms IFT on all benchmarks, demonstrating the advantages of the coarse-grained knowledge obtained through the prior estimation.

**Effect of the posterior estimation** Retaining the cross-entropy loss, we incorporate the KL loss involving the posterior distribution for training. This training method is denoted as Distilling+Posterior. As shown in Figure 3, compared to IFT as well as Distilling+Prior, Distilling+Posterior further boosts the performance. The improvement in performance comes from the posterior distribution capturing more fine-grained knowledge of the closed-source teacher model.

**Combined effect of both** We consider whether combining the KL loss of the prior distribution and the posterior distribution explicitly can improve the performance. Retaining the cross-entropy loss, we directly add the KL loss involving prior distribution and the KL loss involving posterior distribution into the total loss. This training method is denoted as Distilling+Prior+Posterior. As shown in Figure 3, we observe that the performance gain is marginal compared to Distilling+Posterior, with limited improvements seen on only a subset of the benchmarks. The reason for this is that the posterior distribution has already effectively integrated the knowledge from the prior distribution, and the improvement brought by explicitly combining the KL loss terms is limited.

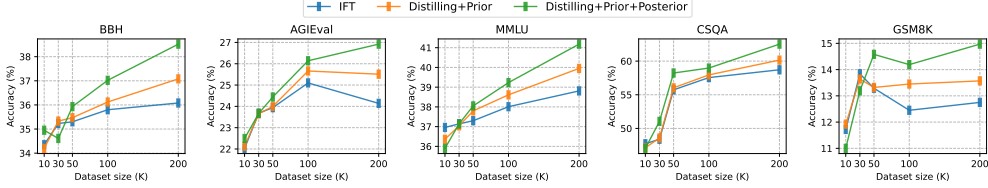

Figure 5: Under different dataset sizes, we investigate the comparison of three methods: IFT, Distilling+Prior, and Distilling+Prior+Posterior, with the student model utilizing LLaMA-7B.

| Models | BBH | AGIEval | MMLU | GSM8K | Average |
|---|---|---|---|---|---|
| GPT-4 (teacher) | 67.4 | 56.4 | 86.4 | 92.0 | 75.5 |
| LLaMA-33B (proxy) | 51.4 | 33.5 | 55.7 | 42.2 | 45.7 |
| LLaMA-13B (proxy) | 42.8 | 26.7 | 45.3 | 20.9 | 33.93 |

Table 5: The performance of closed-source teacher model and aligned proxy models.

| Student Models | Proxy Models | BBH | AGIEval | MMLU | GSM8K | Average |
|---|---|---|---|---|---|---|
| LLaMA-7B | LLaMA-33B | 38.52 | 26.92 | 41.18 | 14.97 | 30.4 |
| LLaMA-7B | LLaMA-13B | 37.41 | 25.67 | 39.56 | 13.83 | 29.12 |

Table 6: Performance of student model with different proxy models.

## 5.3 IMPACT OF SAMPLING ROUNDS

In this section, we discuss the impact of the number of sampling rounds on the posterior estimation. The results are represented in Figure 4. We observe that the best performance is achieved on most benchmarks when the sampling rounds falls within the range of [10,20]. And we find that excessive sampling (e.g., 50 times) results in negative impact on the performance of knowledge distillation. More discussions can be found in Appendix B.2.

## 5.4 IMPACT OF CORPUS SIZE

We investigate the effect of training corpus $\mathcal{C}$ size, as shown in Figure 5. We observe that as the size of the training corpus $\mathcal{C}$ increases, the method "Distilling+Prior+Posterior" consistently outperforms the performance of IFT across benchmarks. A similar trend can also be observed in the method "Distilling+Prior". We analyze that our method benefits from a larger corpus. As the corpus size increases, it becomes more advantageous for the prior estimation to estimate a more accurate and information-rich distribution, subsequently influencing the posterior estimation.

## 5.5 PROXY MODEL SELECTION

Proxy model serves as a bridge between the closed-source teacher model and the open-source student model. And it is first fine-tuned on the corpus generated by the closed-source teacher for preliminary alignment. We believe that opting for a larger and more capable proxy model is advantageous, as it enhances the model's ability to capture the capabilities of the closed-source teacher. Table 5 presents the performance of the proxy models compared to the closed-source teacher. And the student's performance with different proxy models is shown in Table 6. The results validate the advantage of choosing more powerful proxy model.

## 6 CONCLUSION

In this work, we address the challenge of knowledge distillation for closed-source language models, where directly access to the teacher's output distribution is not available. We proposed Bayesian estimation-based knowledge distillation to estimate the output distribution of closed-source language models, achieving superior distillation performance. Our method comprises two main components: prior estimation and posterior estimation. The prior estimation involves obtaining a coarse-grained prior distribution by leveraging the corpus generated by the closed-source language model. The posterior estimation updates prior distribution based on continued sampling results from a proxy model to obtain a fine-grained posterior distribution. Extensive experiments are conducted. The results across various benchmarks consistently show that our method outperforms directly fine-tuning on hard labels, when it comes to knowledge distillation of closed-source language models.

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

| Models | Batch Size | Max Length | Lora Rank | #GPUs | Precision | Dimension | #Heads | #Layers |
|--------|-----------|-----------|-----------|-------|-----------|-----------|--------|---------|
| LLaMA-33B | 1 | 512 | 96 | 8 | float16 | 6656 | 52 | 60 |
| LLaMA-13B | 4 | 512 | 16 | 8 | float16 | 5120 | 40 | 40 |
| LLaMA-7B | 6 | 512 | 16 | 4 | float16 | 4096 | 32 | 32 |

Table 7: Model configurations.

## A  EXPERIMENTAL CONFIGURATIONS

### A.1  TRAINING CONFIGURATIONS

The model configurations are provided in Table 7. We train the student models for three epochs, experimenting with learning rates of 1e-5, 3e-5, and 5e-5 during training. In the knowledge distillation process, we use the following hyperparameters: For the total loss, $\alpha = 0.5$ and $\beta = 1$. For prior estimation, we set $\gamma = 3$ and $n = 5$. For posterior estimation, we conduct 10 rounds of sampling. We evaluate the models on the benchmarks using the final checkpoint. For time efficiency and memory saving, we employ LoRA (Hu et al., 2021) for more efficient training.

### A.2  TRAINING COST

We conducted all our model training on NVIDIA V100 GPUs equipped with 32GB memory. The table 8 presents the GPU and time costs per epoch for various models trained on the OpenOrca dataset. For all student models, we train on the dataset for 3 epochs.

| Models | #GPUs | Hours/Epoch |
|--------|-------|-------------|
| LLaMA-7B | 4 | 17.0 |
| LLaMA-13B | 8 | 15.5 |
| LLaMA-33B | 8 | 40.0 |

Table 8:  The GPU and time costs for various models trained on the 200K OpenOrca dataset.

### A.3  DATA USAGE PER STAGE

Table 9, summarizes the training data used for each model at every stage. Specifically, Orca200K denotes the OpenOrca corpus (Mukherjee et al., 2023) with 200K samples, while Alpaca52K represents the Alpaca corpus (Taori et al., 2023) with 52K samples.

| Models | Prior Estimation Stage | Posterior Estimation Stage | Training Stage |
|--------|-----------------------|----------------------------|----------------|
| LLaMA-7B (IFT) | - | - | Orca200K |
| LLaMA-7B (ours) | Orca200K | Orca200K | Orca200K |
| OpenOrca-Preview1-13B | - | - | Orca200K |
| LLaMA-13B (IFT) | - | - | Orca200K |
| LLaMA-13B (ours) | Orca200K | Orca200K | Orca200K |
| LlaMA-33B (Proxy) | - | - | Orca200K |
| Alpaca-7B | - | - | Alpaca52K |
| LLaMA-7B (ours) | Alpaca52K | Alpaca52K | Alpaca52K |
| Alpaca-13B | - | - | Alpaca52K |
| LLaMA-13B (ours) | Alpaca52K | Alpaca52K | Alpaca52K |
| LlaMA-33B (Proxy) | - | - | Alpaca52K |

Table 9:  Summary of training data for each model at each stage.

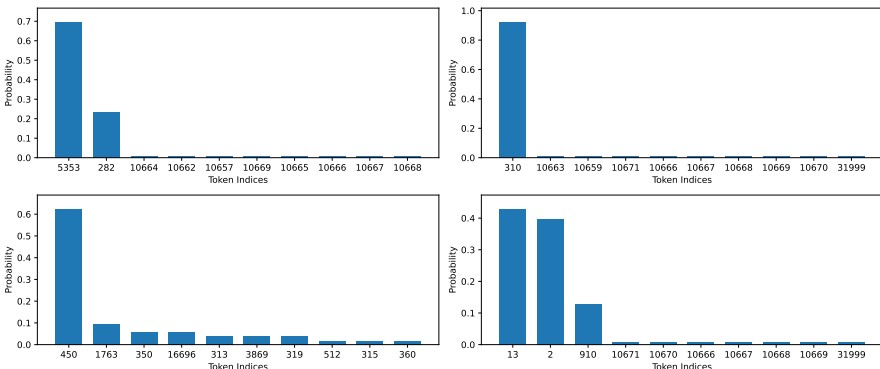

Figure 6: The issue of probability sparsity in the output distribution. A significant portion of probability values concentrates on a few tokens, while the probabilities for other tokens are close to zero.

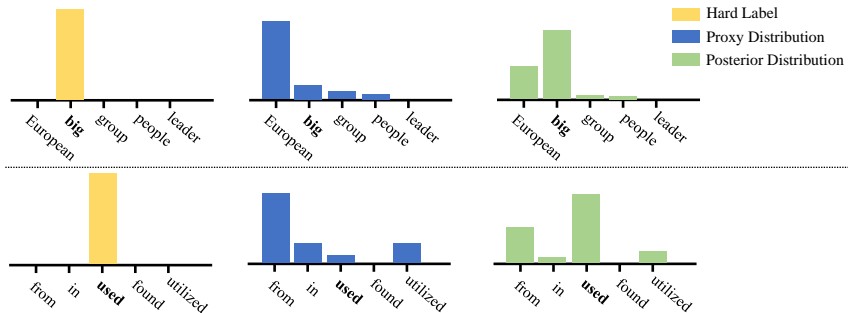

Figure 7: Discrepancies between the the ground-truth distribution and the output distribution of proxy model (proxy distribution) in terms of the top-4 token, while the posterior distribution can stay consistent with the ground-truth distribution.

# B   DISTRIBUTION ANALYSIS

## B.1   PROBABILITY SPARSITY

During the distillation process, we observed a phenomenon of probability sparsity in the output distribution of the proxy model. Typically, only a few tokens have high probabilities, while the probabilities of other tokens are close to zero, as shown in Figure 6. In our distillation process, we retained only the probabilities of the top ten tokens with the highest probabilities, setting the probabilities of the remaining tokens to zero. This phenomena indicates that during the sampling process of the proxy model, we don't need to perform a large number of samples to cover all tokens with non-zero probabilities.

## B.2   DISTRIBUTION DISCREPANCY

We observe that as the number of sampling rounds increased, the model's performance improved on most benchmarks. However, when the number of sampling rounds becomes excessive, such as 50 rounds, the model's performance started to decrease, as shown in Figure 4. We analyze that when the number of sampling rounds becomes excessive, the posterior distribution tends to degenerate into the proxy distribution. When directly using the proxy distribution for knowledge distillation, we observe discrepancies between the proxy distribution and labels generated by teacher (For example, when the label generated by teacher at the current position is "\n", the proxy distribution assigns a high probability (e.g., 0.99) to "<\s>", while the probability of "\n" becomes close to 0.), which can lead to issues in distillation. More cases are shown in Figure 7.

| Tasks | LLaMA-13B (IFT) | LLaMA-13B (ours) | LLaMA-7B (IFT) | LLaMA-7B (ours) |
|---|---|---|---|---|
| Boolean Expressions | 58.8 | 62.4 | 65.06 | 66.4 |
| Causal Judgement | 61.27 | 63.01 | 56.98 | 61.85 |
| Date Understanding | 50.0 | 54.02 | 49.3 | 49.26 |
| Disambiguation QA | 56.8 | 60.0 | 49.4 | 54.8 |
| Formal Fallacies | 56.4 | 54.4 | 54.0 | 54.0 |
| Geometric Shapes | 25.2 | 23.6 | 12.42 | 22.4 |
| Hyperbaton | 63.6 | 66.8 | 49.2 | 54.8 |
| Logical Deduction (5 objects) | 33.8 | 36.14 | 26.51 | 30.96 |
| Logical Deduction (3 objects) | 23.39 | 30.12 | 18.7 | 18.11 |
| Logical Deduction (7 objects) | 44.2 | 51.6 | 42.17 | 42.8 |
| Movie Recommendation | 77.59 | 79.32 | 50.78 | 53.42 |
| Navigate | 51.6 | 56.8 | 45.6 | 55.2 |
| Penguins in a Table | 32.61 | 36.11 | 30.58 | 34.91 |
| Reasoning about Colored Objects | 39.6 | 42.8 | 27.54 | 30.33 |
| Ruin Names | 36.4 | 33.8 | 15.2 | 14.8 |
| Salient Translation Error Detection | 31.6 | 37.2 | 24.0 | 28.4 |
| Snarks | 48.31 | 52.25 | 43.82 | 45.7 |
| Sports Understanding | 60.8 | 60.4 | 56.0 | 55.6 |
| Temporal Sequences | 17.28 | 11.2 | 13.49 | 9.68 |
| Tracking Shuffled Objects (5 objects) | 19.46 | 21.1 | 17.2 | 17.74 |
| Tracking Shuffled Objects (7 objects) | 14.63 | 17.17 | 11.98 | 14.8 |
| Tracking Shuffled Objects (3 objects) | 37.5 | 36.02 | 33.9 | 32.52 |
| Average | 42.77 | **44.83** | 36.08 | **38.52** |

Table 10: Zero-shot performance comparison in Big-Bench Hard benchmark on multiple-choice questions.

## C    EXPERIMENTAL RESULTS

### C.1    DETAILED RESULTS

Following the settings in OpenOrca-Preview1-13B[3] of paper Mukherjee et al. (2023), and considering time efficiency, we conduct training on a subset of the original corpus containing 200k instances. The detailed experimental results for the LLaMA model on BBH, AGIEval, and MMLU benchmarks are presented in Table 10, Table 11 and Table 12.

### C.2    RESULTS OF FLANT5

We also conducted experiments on the FlanT5 (Longpre et al., 2023) model using the OpenOrca dataset, and the results are shown in the Table 13. We find that, compared to the IFT method, our approach does lead to some improvement, although the improvement is limited. We speculate that this might be because FlanT5 is a model that has been fine-tuned with instructions, and its original model already had some basic capabilities for these tasks. Therefore, the additional training results in limited improvement.

### C.3    CONTINUOUS TRAINING OF PROXY MODEL

We also investigated the impact of continuous fine-tuning of the proxy model on the OpenOrca corpus, as shown in the Figure 8. We find that as the number of epochs for fine-tuning the proxy model increases, it leads to a decrease in the performance of posterior estimation. We speculate that this may be due to the proxy model overfitting to the current corpus, resulting in a decrease in the effectiveness. During training, we avoid excessive fine-tuning epochs for the proxy model.

### C.4    ORDER OF N

We investigate the impact of the order of n. Intuitively, the order of n should be selected within a limited range. We conduct experiments distilling on the prior distribution with LLaMA-7B under different order of n, as shown in Table 14.

---

[3] https://huggingface.co/Open-Orca/OpenOrca-Preview1-13B

| Models | #Params | AQuA-RAT | LogiQA | LSAT-AR | LSAT-LR | SAT-English (w/o Psg.) | SAT-Math | Average |
|---|---|---|---|---|---|---|---|---|
| LLaMA-7B (IFT) | 7B | 19.71 | 26.81 | 18.22 | 27.44 | 30.35 | 22.29 | 24.14 |
| LLaMA-7B (ours) | 7B | 22.39 | 29.68 | 19.46 | 33.33 | 30.46 | 26.19 | 26.92 |
| LLaMA-13B (IFT) | 13B | 18.61 | 27.59 | 17.7 | 34.58 | 36.27 | 25.7 | 26.74 |
| LLaMA-13B (ours) | 13B | 25.22 | 29.63 | 19.65 | 36.67 | 33.5 | 31.43 | **29.35** |

Table 11: Performance comparison in AGIEval benchmark on the selected multiple-choice English questions. We use OpenOrca dataset as training corpus.

| Models | #Params | Humanities | Other | Social Sciences | STEM | Average |
|---|---|---|---|---|---|---|
| LLaMA-7B (IFT) | 7B | 38.49 | 44.63 | 40.24 | 31.87 | 38.81 |
| LLaMA-7B (ours) | 7B | 41.4 | 47.32 | 42.17 | 33.82 | 41.18 |
| LLaMA-13B (IFT) | 13B | 46.02 | 53.19 | 48.24 | 33.91 | 45.34 |
| LLaMA-13B (ours) | 13B | 47.81 | 56.7 | 51.36 | 36.79 | **48.17** |

Table 12: Performance comparison on the Massive Multitask Language Understanding benchmark.

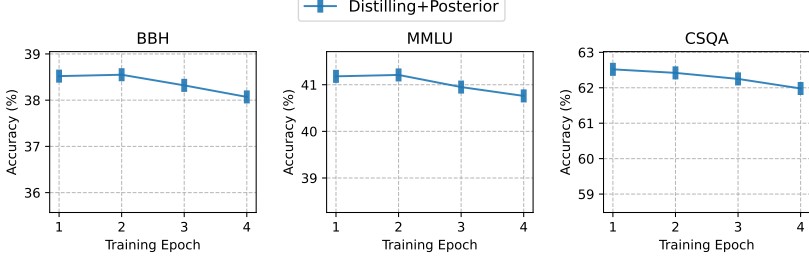

Figure 8: The change in performance of distilling on the posterior distribution (Distilling+Posterior) with the fine-tuning epochs of the proxy model. We utilize LLaMA-7B as the student model, and LLaMA-33B as the proxy model.

| Models | #Params | BBH | AGIEval | ARC | MMLU | CSQA | GSM8K | Average |
|---|---|---|---|---|---|---|---|---|
| GPT-4 | - | - | 56.4 | - | 86.4 | - | 92.0 | - |
| FlanT5-large (IFT) | 780M | 34.63 | 28.12 | 46.44 | 39.41 | 76.78 | 4.54 | 38.32 |
| FlanT5-large (ours) | 780M | 35.22 | 28.84 | 46.61 | 39.34 | 76.93 | 4.71 | **38.61** |
| FlanT5-xl (IFT) | 3B | 38.47 | 28.34 | 59.6 | 46.91 | 84.79 | 6.12 | 44.04 |
| FlanT5-xl (ours) | 3B | 39.51 | 30.1 | 60.12 | 46.78 | 85.38 | 7.1 | **44.83** |

Table 13: The results of the FlanT5 models with different parameter sizes on the six benchmarks. We compare our method with IFT.

| Models | Order of n | BBH | AGIEval | MMLU | GSK8K |
|---|---|---|---|---|---|
| GPT-4 | - | 67.4 | 56.4 | 86.4 | 92.0 |
| LLaMA-7B (IFT) | - | 36.8 | 24.14 | 38.81 | 12.65 |
| LLaMA-7B (ours) | 3 | 37.3 | 25.53 | 40.1 | 13.1 |
| LLaMA-7B (ours) | 5 | 37.3 | 25.7 | 40.0 | 13.2 |
| LLaMA-7B (ours) | 8 | 37.3 | 24.84 | 39.6 | 13.0 |
| LLaMA-7B (ours) | 100 | 36.2 | 24.3 | 38.7 | 12.7 |

Table 14: The results of LLaMA-7B distilled on the prior distribution with different orders of n.

