# OpenReview forum: "Knowledge Distillation for Closed-Source Language Models"
_ICLR.cc/2024/Conference — Submitted to ICLR 2024_

### Official Review · Reviewer_4c71 · 2023-10-31

**Soundness:** 3 good
**Presentation:** 3 good
**Contribution:** 3 good
**Rating:** 6
**Confidence:** 3

**Summary:**

The paper develops a  Bayesian estimation-based knowledge distillation method to alleviate the limitation of traditional knowledge distillation methods.

**Strengths:**

1. The developed Bayesian estimation-based knowledge distillation method is technical sounds and can contribute to improve the performance of knowledge distillation
2. The organize and writing of this paper is good

**Weaknesses:**

1. The main shortcoming is the motivation of technical details is not clear to me, which will also confuse the readers.

**Questions:**

Thanks for your awesome work. The authors propose a Bayesian estimation-based knowledge distillation method to alleviate the limitation of traditional knowledge distillation methods. And I really appreciate this work because the uncertainty plays an important role in language generation and  Bayesian based distribution estimation can be naturely used to solve this problem

I have captured the main idea of this paper, but get lost in the motivation of technical details

1. For prior estimation, I understand it as the authors wanna to obtain the probs of hard lable provided by close-form LLMs. The main question is how the authors to collect the corpus $C$, and will you use the given sequence (w_t, ..., I) as the prompt?

2. For posterior estimation, what is the motivation of this part. I do understand why you update prior here. Moreover, why you process iteratively sampling on open-source LLMs to obtain the posterior distribution of given token? Further, for  open-source LLMs, the probs accross the whole vocabulary can be directly obtained and do need to be calculated like the estimation in Eq.2 and Eq.3

3. Why you use KL as loss function, will it be different from cross entroy loss in normal LLMs training?

---

> ### Author Response · Authors · 2023-11-22
> **Official Comment by Authors**
>
> We sincerely appreciate the time and effort you put into reviewing our paper. And we are grateful that you find our work to be technically sound.
>
> > Q1: How the authors to collect the corpus $\mathcal{C}$, and will you use the given sequence $(w_t,\dots,I)$ as the prompt?
>
> A1: The corpus $\mathcal{C}$ is pre-collected by other works, so we do not collect it ourselves. The corpus $\mathcal{C}$ mainly utilized in this paper is the OpenOrca (Mukherjee et al., 2023) dataset, which itself consists of 1 million samples generated by GPT-4. Specifically, OpenOrca was created by prompting GPT-4 with diverse inputs and collecting the corresponding output sequences. We will incorporate this clarification in the updated version of the paper.
>
> > Q2: For posterior estimation, what is the motivation of this part? I do understand why you update prior here. Why you process iteratively sampling on open-source LLMs to obtain the posterior distribution of given token?
>
> A2: To perform knowledge distillation in traditional approaches, we propose to estimate the output distributions of closed-source LLMs within a Bayesian estimation framework, which includes both prior and posterior estimation. For a specific text input, prior estimation leverages the corpus generated by closed-source language models to derive a coarse-grained approximation for the distribution of the output. Meanwhile, posterior estimation relies on another open-source LLM as a proxy to fine-tune the results of prior estimation. This proxy model serves as a bridge between the teacher (closed-source) and the student (open-source) models, as illustrated in Figure 1. Therefore, the proxy model is selected to be a larger language model than the student model and is initially aligned with the closed-source teacher model using the aforementioned corpus. Finally, we perform knowledge distillation using the estimated output distributions of the closed-source teacher LLM. We will include the above clarification in the updated version of the paper.
>
> > Q3: Further, for open-source LLMs, the probs can be directly obtained and do not need to be calculated like the estimation in Eq.2 and Eq. 3.
>
> A3: We would like emphasize the motivation behind this paper as follows. This paper focuses on enhancing the capabilities of smaller models through knowledge distillation from closed-source language models. However, due to the incapability to directly access the weights, hidden states, and output distributions of these closed-source models, we propose to estimate the output distributions of closed-source language models within a Bayesian estimation framework. By leveraging the estimated output distribution of closed-source language models, traditional knowledge distillation can be executed.
> Therefore, the underlying assumption and premise of this paper is that we are unable to access the prediction distributions of closed-source LLMs.
>
> > Q4: Why you use KL as loss function, will it be different from cross-entropy loss?
>
> A4: In certain cases, the KL loss will differ from the cross-entropy loss. Specifically, when the provided labels are in one-hot form, the KL loss essentially becomes the cross-entropy loss. Whereas, when the labels are represented as a normal distribution rather than in one-hot form, the two loss functions are different and the KL loss is typically employed.
>
> We appreciate the time and effort invested by the reviewer. We hope the above discussions adequately address your concerns.

---

### Official Review · Reviewer_gDYz · 2023-11-01

**Soundness:** 3 good
**Presentation:** 3 good
**Contribution:** 3 good
**Rating:** 6
**Confidence:** 4

**Summary:**

This paper studies knowledge distillation from blackbox large language models to improve zero-shot performance of a student LM. It proposes to estimate the inaccessible output distribution of an LLM and use estimated distributions as the supervision signal for a student LM. It estimates the distribution of a corpus generated by the teacher model. Then it derives a posterior distribution by continued sampling from a proxy of the teacher model. The method appears more effective finetuning the student on teacher-generated corpora and instruction finetuning.

**Strengths:**

- The method aims to improve general-purpose language models and perform zero-shot evaluation on multiple tasks. This is valuable and general.
- The distillation method based on distribution estimation appears effective. Experiments and analyses are reported on several recent datasets.
- Ablation study about method components, sampling rounds, and corpus size is provided.

**Weaknesses:**

- It'd be helpful to include a table that summarizes the training data for each model at each training stage.
- While the results are appealing, there exist key questions to be addressed about why and how the method works. Please refer to Questions.

**Questions:**

1. Is the student trained only on the teacher generated text, or also some ground truth corpora?
2. Is both the cross-entropy loss and the **prior distribution** loss necessary? If we remove $gamma$, then the prior distribution is the same as the teacher-generated corpus distribution. Training on the entire corpus via cross-entropy appears the same as fitting the prior distribution.
3. The $gamma$ term can smooth the distribution. Is the prior estimation step novel or basically the same as label smoothing?
4. Assume the teacher distributions are available in the **posterior estimation** step, and there is no need to learn a proxy model. Will sampling from this teacher to get a posterior distribution help? You could run an experiment where the teacher is open-sourced. Will the posterior distribution be the same as the teacher-generated corpus distribution?
5. Does the improvement from adding the posterior distribution actually due to the ensemble of the teacher and the proxy model?
6. How important is the choice of the proxy model? Can you compare multiple options, present how well they fit the teacher model, their performance on held-out data compared to the teacher model, and how they affect the student performance?

---

> ### Author Response · Authors · 2023-11-22
> **Official Comment by Authors (Part 1)**
>
> We are grateful for the valuable suggestions and questions raised by the reviewer. We hope that the following responses can effectively address your concerns.
>
> > Q1: It’d be helpful to include a table that summarizes the training data for each model at each training stage.
>
> A1: We appreciate your suggestion. In the revised version of our paper, we will include the table below, summarizing the training data used for each model at each stage. Specifically, OpenOrca denotes the OpenOrca corpus (Mukherjee et al., 2023), while Alpaca represents the Alpaca corpus (Taori et al., 2023). Thank you for bringing attention to this important aspect.
>
> | Models | Prior Estimation Stage | Posterior Estimation Stage | Training Stage |
> | ---- | :----: | :----: | :----: |
> | LLaMA-7B (IFT) (Table 2) | - | - | OpenOrca |
> | LLaMA-7B (IFT) (Table 2) | OpenOrca | OpenOrca | OpenOrca |
> | OpenOrca-Preview1-13B (Table 2) | - | - | OpenOrca |
> | LLaMA-13B (IFT) (Table 2) | - | - | OpenOrca |
> | LLaMA-13B (ours) (Table 2) | OpenOrca  | OpenOrca  | OpenOrca |
> | LLaMA-33B (Proxy Model) (Table 2) | - | - | OpenOrca |
> | Alpaca-7B (Table 3) | - | - | Alpaca|
> | LLaMA-7B (ours) (Table 3) | Alpaca| Alpaca| Alpaca |
> | Alpaca-13B (Table 3) | - | - | Alpaca|
> | LLaMA-7B (ours) (Table 3) | Alpaca| Alpaca| Alpaca |
> | LLaMA-33B (Proxy Model) (Table 3) | - | - | Alpaca|
>
> > Q2: Is the student trained only on the teacher generated text, or also some ground truth corpora?
>
> A2: In our experiment, all the student models and the proxy model are exclusively trained on the text generated by the closed-source teacher model. If a ground truth corpus is available, the loss between the ground truth labels and the student's predictions can be incorporated into the total loss. We will clarify this in the revised paper.
>
> > Q3: Is both the cross-entropy loss and the prior distribution loss necessary? If we remove $\gamma$, then the prior distribution is the same as the teacher-generated corpus distribution. Training on the entire corpus via cross-entropy appears the same as fitting the prior distribution.
>
> A3: We would like to clarify that the prior distribution is distinct from the teacher-generated corpus distribution. The former is an initially estimated distribution of the closed-source LLM's output, while the latter refers to the output sequence in the form of hard labels.
>
> > Q4: Is the prior estimation step novel or basically the same as label smoothing?
>
> A4: To perform knowledge distillation in traditional approaches, we propose to estimate the output distributions of closed-source LLMs within a Bayesian estimation framework, which includes both prior and posterior estimation. For a specific text input, prior estimation leverages the corpus generated by closed-source language models to derive an initial approximation for the distribution of the output. Therefore, we can note that prior estimation draws inspiration from traditional n-gram language modeling, with label smoothing serving as just a component addressing the challenges associated with zero probabilities, particularly as n becomes large, within this comprehensive framework. Therefore, it is crucial to recognize that prior estimation differs from label smoothing in both their forms and purposes. We will clarify this in the updated paper.
>
> > Q5: Assume the teacher distributions are available in the posterior estimation step. Will sampling from this teacher to get a posterior distribution help? Will the posterior distribution be the same as the teacher-generated corpus distribution?
>
> A5: We address this question from three perspectives.
> - Firstly, in theory, we can prompt the closed-source teacher model to sample the next token. However, due to the high cost of API calls, in practice, we resort to the use of proxy model instead.
> - Secondly, given that the primary motivation of this paper is to estimate the output distributions of closed-source LLMs (teacher) within a Bayesian estimation framework, the need for conducting posterior estimation would be obviated if the teacher distributions are readily available.
> - Thirdly, posterior estimation relies on the proxy model to refine the results of prior estimation. This proxy model functions as a crucial link between the teacher (closed-source) and the student (open-source) models. If the teacher distributions were indeed available, it would undoubtedly be preferable to sample from this distribution rather than relying on the proxy model's distribution.

---

> ### Author Response · Authors · 2023-11-22
> **Official Comment by Authors (Part 2)**
>
> > Q6: Does the improvement from adding the posterior distribution actually due to the ensemble of the teacher and the proxy model?
>
> A6: The observed improvement could be attributed to some ensemble effect, yet there are fundamental distinctions.
> - Firstly, ensemble methods necessitate the execution of multiple models during inference, whereas knowledge distillation only requires the application of the student model.
> - Secondly, although the student model learns from both the closed-source teacher model and the proxy model, akin to multi-teacher knowledge distillation, the primary purpose of the proxy model is to act as a bridge between the closed-source teacher and the open-source student. This underscores the significance of a pre-alignment between the proxy and the teacher models.
>
> > Q7: The choice of the proxy model. Compare multiple options, present how well they fit the teacher model, and their performance on held-out data, and how they affect the student performance.
>
> A7: Thank you for your valuable suggestion. We believe that opting for a larger and more capable proxy model is advantageous, as it enhances the model's ability to capture the capabilities of the closed-source LLM. Taking into consideration this factor, along with the constraints posed by computational resources and time limitations, we have selected LLaMA-33B as the proxy model for our experiments. In response to the raised question, we have also explored alternative options.
>
> The table below presents the performance of both the closed-source teacher (GPT-4) and the proxy models (LLaMA-33B and LLaMA-13B):
>
> | Models | BBH | AGIEval | MMLU | GSM8K |
> | ---- | :----: | :----: | :----: | :----: |
> | GPT-4 (teacher) | 67.4 | 56.4 | 86.4 | 92.0 |
> | LLaMA-33B (proxy) | **51.7** | **33.5** | **55.7** | **42.2** |
> | LLaMA-13B (proxy) | 42.8 | 26.7 | 45.3 | 20.9 |
>
> Furthermore, we employed LLaMA-7B as the student model and compared its performance with different proxy models (LLaMA-33B and LLaMA-13B) in the table below:
>
> | Models | Proxy Models | BBH | AGIEval | MMLU | GSM8K |
> | ---- | ---- | :----: | :----: | :----: | :----: |
> | LLaMA-7B (student) | LLaMA-33B (proxy) | **38.52** | **26.92** | **41.18** | **14.97** |
> | LLaMA-7B (student) | LLaMA-13B (proxy) | 37.41 | 25.67 | 39.56 | 13.83 |
>
> The results validate the earlier assumption. We will incorporate these new experiments into the updated version of the paper.

---

### Official Review · Reviewer_Ryd9 · 2023-11-02

**Soundness:** 3 good
**Presentation:** 3 good
**Contribution:** 2 fair
**Rating:** 5
**Confidence:** 5

**Summary:**

The paper proposes a knowledge distillation method for closed-source large language models, such as distilling GPT to other public LLMs.

**Strengths:**

- The paper is easy to read and follow.

**Weaknesses:**

- The motivation is not significant enough. If we would like to compress our large models, we usually can directly access the prediction distribution. It's unsure whether distilling other closed-source models is a reasonable research topic, which often violates the usage policies of closed-source LLMs.

- The performance gain seems marginal compared with baselines.

- Only 7B/13B pre-trained models are tuned in the experiments.

- The cost is not discussed in the paper.

**Questions:**

- The motivation is not significant enough. If we would like to compress our large models, we usually can directly access the prediction distribution. It's unsure whether distilling other closed-source models is a reasonable research topic, which often violates the usage policies of closed-source LLMs. More explanations about motivation can be discussed.

- The performance gain seems marginal compared with baselines. The significance of the method can be better described.

- Only 7B/13B pre-trained models are tuned in the experiments. Larger-size models can be added in the experiments.

- Only pre-trained models are tuned in the experiments. Can we use the method to train the models from scratch?

- The cost can be discussed in the paper.

---

> ### Author Response · Authors · 2023-11-22
> **Official Comment by Authors**
>
> We are grateful that the reviewer find our paper to be a good read. And we appreciate the time and effort the reviewer put into reviewing our paper. We hope that the following responses can address your concerns.
>
> > Q1: The motivation is not significant enough. It’s unsure whether distilling other closed-source models is a reasonable research topic, which often violates the usage policies of closed-source LLMs.
>
> A1: We would like to emphasize that the motivation behind this paper is both clear and significant. In recent months, there has been a trend focused on transferring the capabilities of closed-source LLMs like GPT-4 and GPT-3.5 into open-source models. Notable studies, including Alpaca (Taori et al., 2023), Vicuna (Chiang et al., 2023), WizardLM (Xu et al., 2023), and OpenOrca (Mukherjee et al., 2023), have dedicated efforts toward this objective, primarily through fine-tuning on samples generated by closed-source LLMs. However, these approaches only enable open-source language models to minic the input-output behavior of closed-source LLMs without acquiring their intrinsic knowledge related to logits, weights, activations, and other relevant aspects. In contrast, we propose to estimate the output distributions of closed-source LLMs within a Bayesian estimation framework and perform traditional knowledge distillation.
>
> > Q2: The performance gain seems marginal compared with baselines.
>
> A2: Indeed, within the domain of knowledge distillation, previous works like TinyBert (Jiao et al., 2020), PKD (Sun et al., 2019), BERT-of-Theseus (Xu et al., 2020), and MGSKD (Liu et al., 2022) have typically shown an average performance gain of approximately 1-2 points. In contrast, our main results, as depicted in Table 2, indicate a more significant average performance gain of about 2-3 points over the baseline models. This discrepancy is noteworthy, underscoring a non-marginal improvement in the field of knowledge distillation.
>
> > Q3: The cost is not discussed.
>
> A3:  Thanks for the suggestion. We conducted all our model training on NVIDIA V100 GPUs equipped with 32GB memory. The table below presents the GPU and time costs per epoch for various models trained on the OpenOrca dataset. This detailed information will be included in the revised version of our paper.
>
> | Models | #GPUs | Hours/Epoch |
> | ---- | :----: | :----: |
> | LLaMA-7B | 4 | 17.0 |
> | LLaMA-13B | 8 | 15.5 |
> | LLaMA-33B | 8 | 40.0 |
>
> > Q4: Only 7B/13B are tuned. Larger-size models can be added.
>
> A4: We would like to clarify this question as follows. The primary goal of knowledge distillation is to transfer knowledge from large teacher models to smaller student models. Additionally, the choice of the proxy model as LLaMA-33B is made to serve as a bridge between the closed-source teacher model and the student model, thereby limiting our options in terms of the size of the student model. Consequently, we opted for the 7B and 13B LLaMA as our student models.
>
> > Q5: Can we use the method to train the models from scratch?
>
> A5: In theory, it is feasible to train the student model from scratch using the proposed method, akin to other knowledge distillation approaches. However, practical considerations arise since pre-training corpora often differ in size and domains from knowledge distillation datasets, and these disparities may introduce challenges. Consequently, knowledge distillation commonly relies on a pre-trained student model.
>
> We once again thanks the valuable suggestions and questions raised by the reviewer.

---

> > ### Comment · Reviewer_Ryd9 · 2023-12-05
> >
> > Thank you for the clarifications. I've read other reviews and rebuttals.

---

### Official Review · Reviewer_QK8L · 2023-11-09

**Soundness:** 2 fair
**Presentation:** 2 fair
**Contribution:** 3 good
**Rating:** 6
**Confidence:** 3

**Summary:**

When we apply knowledge distillation to a closed-source language model, we cannot access probability for each token in the vocabulary. This paper proposes ways to estimate these probabilities. Specifically, "prior" probability can be estimated via fitting an n-gram on the generated corpus (by the closed-source language model). Then the authors further propose to estimate "posterior" probability by sampling a proxy open-source language model. The distillation loss is a combination of KL divergences between the token probabilities of student model, and 1) one-hot target; 2) "prior"; 3) "posterior". Experiments suggest adding 2 and 3 is beneficial.

**Strengths:**

Overall, the method seems novel. The probabilities of tokens generated from teacher model is crucial for knowledge distillation. The paper proposes some interesting techniques to estimate it when the teacher model is a blackbox.

**Weaknesses:**

A major concern is clarity. Section 3.2.1 and 3.2.2 are not clearly written. In Eq (4), what is the exact form of $f_{W_t}(\cdot)$. Morever, after reading section 3.2.2, it is not clear to me how exactly the "posterior" is updated, especially when the form of $f_{W_t}(\cdot)$ is not given.
See more weakness in questions below.

**Questions:**

1. The author uses n-gram on teacher generated corpus to estimate the "prior". How robust is the result w.r.t. the order $n$? How big is the generated corpus? Are there many $w_t$ such that #$(w_t, w_{t-1}^\prime, \dots, w_{t-n}^\prime)=0$?  Is it necessary to apply smoothing? Also, according to Eq. (2) and (3), $\sum_{w_t}p_{w_t}\neq 1$, which seems unusual.

2. Estimating the "posterior" relies on a proxy model. A relevant stream of work is using teacher assistant models. See for example, https://arxiv.org/pdf/1902.03393.pdf . The authors should at least discuss the connection.

---

> ### Author Response · Authors · 2023-11-22
> **Official Comment by Authors**
>
> We sincerely thank you for the time and effort you have put into reviewing our paper. We appreciate your insightful feedback and are pleased that you find our work to be novel.
>
> > Q1: It is not clear to me how exactly the “posterior” is updated, especially when the form of $f_{W_t}$ is not given.
>
> A1: The general idea of posterior estimation is to utilize the output distributions of the proxy model for calibrating the coarse-grained prior distribution. The process of posterior updating is to update according to the conditional probability formula, such as calculating the likelihood and prior, etc. It will be easier to understand with a discrete case example. To explain the process more clearly, we have reorganized this section and will incorporate the updates in the revised version of the paper.
>
> > Q2: How robust is the results w.r.t. the order n?
>
> A2: Intuitively, the order of n should be selected within a limited range. To confirm this, we conduct experiments distilling on the prior distribution with LLaMA-7B under different order of n, as shown in the table below:
>
> | Models | Order of n | BBH | AGIEval | MMLU | GSM8K |
> | ---- | :----: | :----: | :----: | :----: | :----: |
> | GPT-4 (teacher) | - | 67.4 | 56.4 | 86.4 | 92.0 |
> | LLaMA-7B (IFT) | - | 36.8 | 24.14 | 38.81 | 12.65 |
> | LLaMA-7B (ours) | 3 | **37.3** | 25.53 | **40.1** | **13.1** |
> | LLaMA-7B (ours) | 5 | **37.3** | **25.7** | **40.1** | **13.1** |
> | LLaMA-7B (ours) | 8 | **37.3** | 24.48 | 39.6 | 13.0 |
> | LLaMA-7B (ours) | 100 | 36.2 | 24.3 | 38.79 | 12.7 |
>
> The results indicate that n should not be too large. Empirically, we set n=5 in this study. We will also conduct experiments on the posterior distribution distillation and will incorporate the results when updating the paper.
>
> > Q3: How big is the generated corpus?
>
> A3: The generated corpus utilized in this paper comprises 200,000 samples randomly selected from the original OpenOrca (Mukherjee et al., 2023) dataset, which itself contains 1 million samples generated by GPT-4. We will incorporate this clarification in the updated version of the paper.
>
> > Q4: Are there many $w_t$ such that $\\#(w_t,w_{t-1}', ..., w_{t-n}')=0$?
>
> A4: Indeed, that is a valid concern with the n-gram method, particularly when n becomes large. To address this, the order of n is set to 5 to mitigate such cases. Additionally, it's important to note that the prior distribution serves as a coarse-grained estimate, acknowledging scenarios where $\\#(w_t,w_{t-1}',\dots,w_{t-n}')=0$, obviating the need of label smoothing. For a more fine-grained estimate, posterior estimation is employed.
>
> > Q5: According to Eq. 2 and 3, $\sum_{w_t}{p_{w_t}}\neq 1$.
>
> A5: Thanks for your concern. We would like to demonstrate that $\sum_{w_t\in \mathbb{V}}p_{w_t}=1$. We first ensure $\sum_{w_t\in \mathbb{V}}\\#(w_t, w_{t-1}', \dots, w_{t-n}')=\\#(w_{t-1}', \dots, w_{t-n}')$. Then, according to Eq. 2 and Eq. 3, we have:
>
> \begin{equation}
> \sum_{w_t\in \mathbb{V}}p_{w_t} = \frac{\sum_{w_t\in \mathbb{V}, w_t\neq w_t'}{\\#(w_t, w_{t-1}', \dots, w_{t-n}')}}{\gamma\\#(w_{t-1}', \dots, w_{t-n}')} + \frac{\\#(w_t', w_{t-1}', \dots, w_{t-n}')}{\gamma\\#(w_{t-1}', \dots, w_{t-n}')} + \frac{\gamma-1}{\gamma}
>           = \frac{\sum_{w_t\in \mathbb{V}}{\\#(w_t, w_{t-1}', \dots, w_{t-n}')}}{\gamma\\#(w_{t-1}', \dots, w_{t-n}')} + \frac{\gamma-1}{\gamma}
>          = \frac{1}{\gamma} + \frac{\gamma-1}{\gamma} = 1
> \end{equation}
>
> > Q6: A relevant stream of work is using teacher assistant models. The authors should at least discuss the connection.
>
> A6: Indeed, the proxy model serves a role akin to teacher assistant models (Mirzadeh et al., 2019). Our work shares similarities with theirs in two aspects: (1) both address the challenge of knowledge distillation, and (2) both introduce an intermediate network. However, there are notable distinctions.
> - Firstly, teacher assistant models aim to bridge the gap in model sizes between the teacher and the student, whereas the proxy model in this work seeks to facilitate the distillation of closed-source language models. Hence, pre-alignment between the proxy and the teacher models is deemed necessary.
> - Secondly, the proxy model operates by calibrating the prior distribution obtained from a generated corpus by closed-source language models.
>
> We are grateful for the valuable suggestions and questions raised by the reviewer. We hope the above points adequately address each of your concerns.

---

### Author Response · Authors · 2023-11-23
**Looking Forward to the Opportunity for Further Discussion**

Dear Reviewers,

We sincerely appreciate the time and effort you've devoted to reviewing our work. We have endeavored to systematically address the reviewers' questions by providing detailed clarifications and conducting supplementary experiments. Regrettably, we have not received any feedback from you and are currently unable to continue the discussions. Given the impending deadline, we kindly request your attention to our responses. We look forward to the opportunity for further discussions. Thank you for your consideration.

Best regards,

The Authors

---

### Meta-Review · Area_Chair_Pjfu · 2023-12-05

**Metareview:**

This paper proposes an approach for learning a smaller model from a larger teacher model (i.e., knowledge distillation). Unlike standard approaches which simply learn on the one-hot labels from the teacher, this paper presents a method that trains a proxy model on the teacher-generated corpus, which is used to supervise the student model.

How the best learn a performant smaller model from closed-source models is an important problem, and this paper presents an interesting method with some empirical gains. However, there were several issues raised with regard to paper clarity, and whether the improvements in performance was "enough" to justify the added complexity.

**Justification For Why Not Higher Score:**

It is unclear that the problem being addressed here is actual a problem. Moreover, the proposed approach requires sampling from a closed source model, which is generally considered expensive.

**Justification For Why Not Lower Score:**

N/A

---

### Decision · Program_Chairs · 2024-01-16

Reject